# Population sequencing enhances understanding of tea plant evolution

Xinchao Wang [1,6], Hu Feng [2,6], Yuxiao Chang [2,6], Chunlei Ma [1,6], Liyuan Wang [1,6], Xinyuan Hao [1,6], A'lun Li [2], Hao Cheng [1], Lu Wang [1], Peng Cui [2], Jiqiang Jin [1], Xiaobo Wang [2], Kang Wei [1], Cheng Ai[2], Sheng Zhao [2], Zhichao Wu [2], Youyong Li[3], Benying Liu [3], Guo-Dong Wang [4,5✉], Liang Chen [1✉], Jue Ruan [2✉] & Yajun Yang [1✉]

Tea is an economically important plant characterized by a large genome, high heterozygosity, and high species diversity. In this study, we assemble a 3.26-Gb high-quality chromosome-scale genome for the 'Longjing 43' cultivar of *Camellia sinensis* var. *sinensis*. Genomic resequencing of 139 tea accessions from around the world is used to investigate the evolution and phylogenetic relationships of tea accessions. We find that hybridization has increased the heterozygosity and wide-ranging gene flow among tea populations with the spread of tea cultivation. Population genetic and transcriptomic analyses reveal that during domestication, selection for disease resistance and flavor in *C. sinensis* var. *sinensis* populations has been stronger than that in *C. sinensis* var. *assamica* populations. This study provides resources for marker-assisted breeding of tea and sets the foundation for further research on tea genetics and evolution.

[1] Key Laboratory of Tea Biology and Resource Utilization, Ministry of Agriculture and Rural Affairs, National Center for Tea Plant Improvement, Tea Research Institute, Chinese Academy of Agricultural Sciences, 310008 Hangzhou, China. [2] Lingnan Guangdong Laboratory of Modern Agriculture, Genome Analysis Laboratory of the Ministry of Agriculture and Rural Affairs, Agricultural Genomics Institute at Shenzhen, Chinese Academy of Agricultural Sciences, 518120 Shenzhen, China. [3] Tea Research Institute, Yunnan Academy of Agricultural Sciences, 650231 Menghai, China. [4] State Key Laboratory of Genetic Resources and Evolution, Kunming Institute of Zoology, Chinese Academy of Sciences, 650223 Kunming, China. [5] Center for Excellence in Animal Evolution and Genetics, Chinese Academy of Sciences, 650223 Kunming, China. [6] These authors contributed equally: Xinchao Wang, Hu Feng, Yuxiao Chang, Chunlei Ma, Liyuan Wang, Xinyuan Hao. ✉email: wanggd@mail.kiz.ac.cn; liangchen@tricaas.com; ruanjue@caas.cn; yjyang@tricaas.com

Tea [*Camellia sinensis* (L.) O. Kuntze, $2n = 30$] is one of the most important and traditional economic crops in many developing countries in Asia, Africa, and Latin America, and is consumed as a beverage by more than two-thirds of the world's population[1,2]. Originally, tea was used as a medicinal herb in ancient China, and it was not until the Tang dynasty (A.D. 618–907) that it gained popularity as a beverage[3,4]. Since then, tea planting has expanded throughout the world, being notably associated with the influence of trading along the Silk and Tea Horse Roads[5,6]. Subsequent to its initial domestication, tea was further bred and cultivated to enhance certain organoleptic traits, primarily taste, and aroma, as well as biotic and abiotic stress resistance properties, including cold and disease resistance[7]. However, the genes underlying the traits that were gradually selected and expanded remain largely undetermined.

The majority of cultivated tea plants belong to the genus *Camellia* L., section *Thea* (L.) Dyer, in family Theaceae and are categorized as one of two main varieties: *C. sinensis* var. *sinensis* (CSS) and *C. sinensis* var. *assamica* (Masters) Chang (CSA). Of these, CSS is characterized by smaller leaves, cold tolerance, and a shrub or semishrub growth habit, whereas CSA has larger leaves and an arborous or a semiarborous habit[8,9]. Moreover, some *C. sinensis*-related species (CSR) belonging to section *Thea*, such as *Camellia. taliensis* (W.W. Smith) Melchior, *C. crassicolumna* Chang, *C. gymnogyna* Chang, and *C. tachangensis* F.C. Zhang, are locally consumed as tea by inhabitants in certain regions of the Indo–China Peninsula, particularly in Yunnan Province, China. Theoretically, different species are assumed to have experienced reproductive isolation; however, different tea species can readily hybridize, and it is thus difficult to accurately classify the offspring of different hybrids. Moreover, numerous morphological features are continuous, which makes it difficult to identify taxonomic groups[10]. The classification of tea has traditionally been based on morphology, the outcome of which is sometimes found to be inconsistent with the more recent classifications based on molecular characterization[11–15]. However, given that tea plant taxonomy generally lacks comprehensive genomic evidence, further analyses using population resequencing are required to optimize taxonomic assignments at the whole-genome level.

To gain a better understanding of the domestication, breeding, and classification of tea, we collected and sequenced samples of 139 tea accessions from across the world. High-quality annotated genes and chromosome-scale tea genomes are necessary for population research, and in this regard, the previously sequenced genomes of the tea cultivars Yunkang 10 (YK10, CSA)[1] and Shuchazao (SCZ, CSS)[16] are considered important milestones in tea genetic research. However, these two genomes were not characterized at the chromosome scale, and the associated scaffold N50 values were less than 1.4 Mb, thereby precluding an evaluation of phenotypic variation and genome evolution in important intergenic regions. Moreover, annotation of the core genes (Benchmarking Universal Single-Copy Orthologs[17], BUSCO) in the SCZ and YK10 genomes is only 80.58% and 68.58% complete, respectively, and this has tended to hamper further population selection analysis, functional genomic analysis, and molecular breeding research. Therefore, for the purposes of de novo genome assembly in the present study, we focused on the 'Longjing 43' (LJ43) cultivar of *C. sinensis*, which is among the most widely cultivated tea cultivars in China and is characterized by a number of desirable traits, including high cold resistance, extensive plantation adaptation, early sprouting, excellent taste, and a favorable aroma[18].

Herein, we describe the sequencing, assembly, and analysis of a high-quality chromosome-scale tea genome, along with the directions of divergent selection in the CSS and CSA populations, and conduct phylogenetic analysis of tea accessions. However, details regarding the origin of tea and the subsequent routes of expansion remain to be clarified.

## Results

**Sequencing and assembly of the LJ43 genome.** The predicted size of the LJ43 genome was approximately 3.32 Gb (Supplementary Figs. 1 and 2), making it larger than the assembled YK10 (2.90–3.10 Gb)[1] and SCZ (~2.98 Gb)[16] genomes. To enhance genome assembly, 196-Gb PacBio single molecule real-time (SMRT) long reads (Supplementary Table 1) were initially assembled using WTDBG[19] (version 1.2.8; Supplementary Note 1), which resulted in a 3.26-Gb assembled genome containing 37,600 contigs and covering approximately 98.19% of the whole genome. To further enhance the integrity of the assembled genome, contigs were scaffolded based on chromosome conformation capture sequencing (Hi–C) (Supplementary Table 1; Supplementary Figs. 3 and 4; Supplementary Note 1), and the final 3.26-Gb assembly was generated with a scaffold N50 value of 144 Mb. Of the 37,600 initially assembled contigs, 7,071 (~2.31 Gb, 70.9% of the original assembly) were subsequently anchored with orientation into 15 chromosomal linkage groups (Fig. 1b; Supplementary Fig. 5; Supplementary Tables 2 and 3).

To evaluate the quality of the assembled LJ43 genome, we estimated sequence accuracy at both the single-base and scaffold levels. The percentages of homogeneous single-nucleotide polymorphisms (SNPs) and homogeneous insertions-deletions (InDels) in the genome were 0.000224% and 0.000568%, respectively, thereby indicating a low error rate at the single-base level (Supplementary Table 4). The accuracy of the scaffolding was evaluated based on three strategies. First, 5879 (83.14%) of 7071 connections in the Hi–C scaffolds were confirmed with at least two 10× Genomics Chromium linked reads spanning the connections. Second, 5374 (76.00%) connections were confirmed by at least two BioNano Genomics (BNG) optical molecules, among which 4484 (63.41%) overlapped with the connections confirmed by the 10× Genomics Chromium linked reads. In total, 6769 (95.73%) connections in the scaffold generated using Hi–C could be confirmed by 10× Genomics Chromium linked reads or BNG optical molecules, indicating that the scaffold was accurate. Third, the collinearity of the tea genetic map[20] with 3483 single sequence repeat (SSR) markers and the LJ43 genome had a mean coefficient of determination ($R^2$) of 0.93, with maximum and minimum values of 0.98 and 0.84, respectively (Fig. 1c; Supplementary Table 3). In summary, at both the single-base and scaffold levels, the assembly accuracy of the LJ43 genome was high.

**Genome annotation.** For genome annotation, we annotated the repetitive sequences of the genome by combining de novo and homology-based prediction. We identified and masked 2.38 Gb (80.06%) of the LJ43 genome as repetitive sequences (Supplementary Table 5). Among the integrated results, 60.77% (1.98 Gb) of the sequences were long terminal repeat (LTR) retrotransposons (Supplementary Table 6), with LTR/Gypsy elements being the dominant class (49.85% of the whole genome, 1.63 Gb), followed by LTR/Copia elements (7.09%, 231.27 Mb). Comparisons with previously sequenced tea genomes indicated that the LTR/Gypsy and LTR/Copia repeats in the LJ43 genome were similar to those in the SCZ (Gypsy 46%, Copia 8%)[16] and YK10 (Gypsy 47%, Copia 8%)[1] genomes, whereas the LTR/Gypsy and LTR/Copia repeats in tea were expanded compared with those in kiwifruit (*Actinidia chinensis*) (13.4%)[21], silver birch (*Betula pendula*) (10.8%)[22], and durian (*Durio zibethinus*) (29.4%)[23] but contracted compared with those in maize (*Zea mays*) (74.20%)[24].

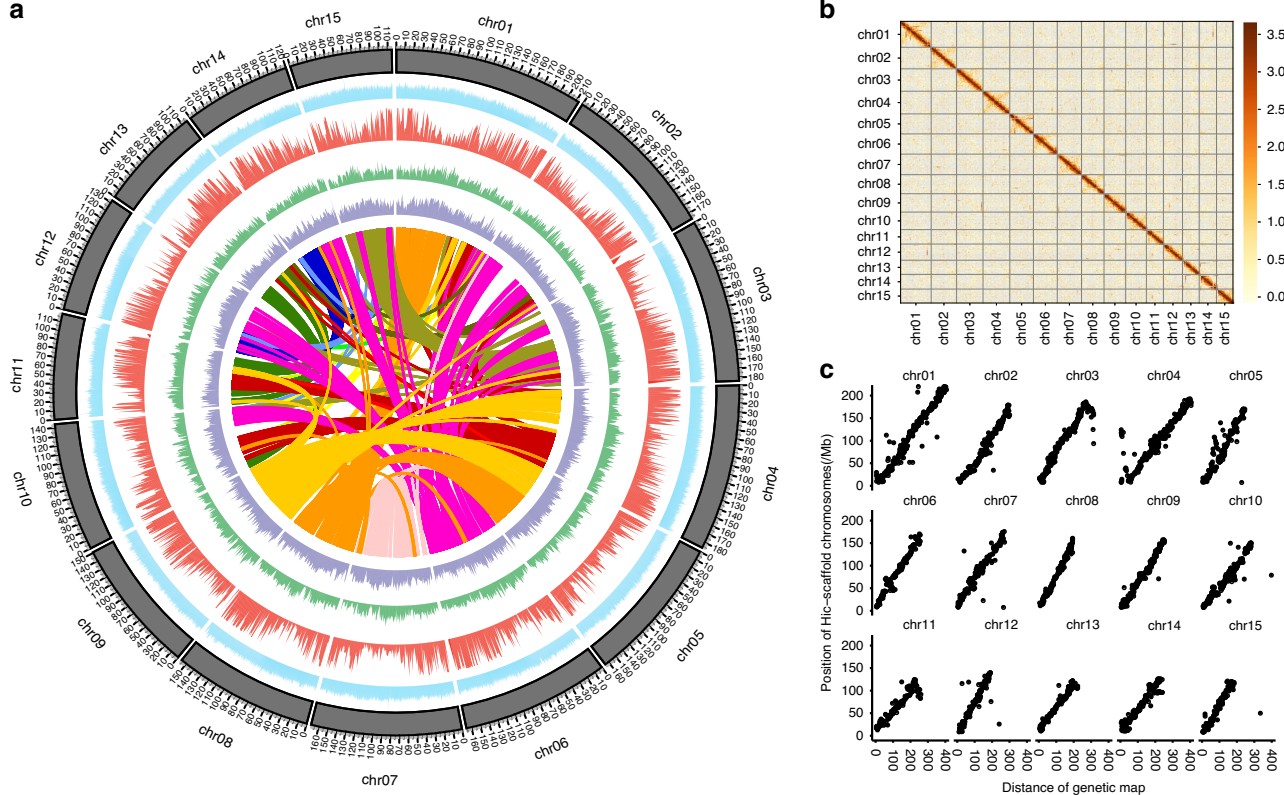

**Fig. 1 Characterization and quality of the LJ43 genome. a** The landscape of the LJ43 genome. From inside to outside: LJ43 gene collinearity; long terminal repeat density (purple); single-nucleotide polymorphism density (green); gene density (red); GC content (blue). The chromosome units of all aforementioned features are 1 Mbp. **b** Genome-wide all-by-all Hi–C interaction. The resolution is 0.5 Mbp. **c** The collinearity of the genetic map and assembled genome. Source data underlying Fig. 1a and c are provided as a Source Data file.

LTR retrotransposons are the predominant repeat elements that tend to be poorly assembled in draft genomes[25], and it has been reported that the LTR assembly index (LAI), which approximates the ratio of intact LTRs to total LTRs, can be exploited to evaluate assembly continuity. Thus, we investigated the LTR composition of the LJ43 genome and compared it with the composition of the SCZ and YK10 genomes and found that the LAI of the LJ43, SCZ, and YK10 genomes was 11.78, 5.45, and 2.86, respectively, thereby indicating that a larger number of intact LTR retrotransposons had been assembled in the LJ43 genome. We subsequently used LTR-finder to detect intact LTR retrotransposons in the three tea genomes, aligned the 5′ and 3′ terminal repeats using MUSCLE (version 3.8.31), and calculated the Kimura two-parameter distance for each alignment using EMBOSS (version 6.4.0). To calculate the insertion time of each LTR, we used the equation Time = Ks/2μ (μ = 6.5E-9)[26]. Unexpectedly, we found that the LTR of LJ43 had accumulated fewer point mutations and obtained calculated peaks of LTR insertion in LJ43, SCZ, and YK10 of 1, 9, and 9 million years ago (mya), respectively (Supplementary Fig. 6). To further investigate this seemingly anomalous pattern, we performed NGS read error correction during genome assembly. Comparison of the genome sequences corrected by PacBio reads and NGS reads revealed that 98.19% of the 5′ and 3′ terminal IR sequences were corrected by NGS reads of no more than three bases (Supplementary Fig. 6d). Moreover, error correction did not alter the Ks, which ranged from 0.013 (the peak of LJ43) to 0.117 (the peak of SCZ and YK10). Taken together, our analyses indicate that the assembled LJ43 genome is more complete than previously sequenced tea genomes, has a high LAI, and contains more recently derived LTRs, which results in

| Table 1 Genome assembly and annotated genes of the three tea cultivars. | | | |
|---|---|---|---|
| | **LJ43** | **SCZ** | **YK10** |
| Genome size (Gb) | 3.26 | 3.14 | 3.02 |
| Contig N50 (kb) | 271.33 | 67.01 | 19.96 |
| Scaffold N50 (Mb) | 143.85 | 1.39 | 0.45 |
| GC content (%) | 38.67 | 37.84 | 39.62 |
| Number of genes | 33,556 | 33,932 | 36,951 |
| Number of exons | 188,681 | 191,870 | 176,616 |
| Length of exons (Mb) | 40.4 | 45.6 | 41.6 |
| Average length of exons (bp) | 226.1 | 237.8 | 235.6 |
| Average length of genes (intron + exon) (bp) | 10,815.5 | 7385 | 3548 |
| Average number of exons per gene | 5.3 | 5.7 | 4.8 |
| Average coding sequence length (bp) | 1205 | 1345 | 1131 |
| BUSCO (%) | 88.36 | 80.58 | 68.58 |

contrasting estimates of LTR insertion time among LJ43, SCZ, and YK10.

To assist in gene prediction, we generated a total of 340 Gb of clean RNA-seq data from 19 samples of five tissue types (bud, leaf, flower, stem, and root) collected in each of the four seasons (with the exception of flowers during summer; Supplementary Table 7). A total of 33,556 protein-coding genes with an RNA-seq coverage ratio greater than 50% were annotated, with an average gene size of 10,816 bp (Supplementary Note 2; Supplementary Fig. 7) and a mean number of 5.3 exons per gene (Table 1). Subsequently, we assessed LJ43 genome annotation integrity using the BUSCO database[17] and found that 1215 (88.36%)

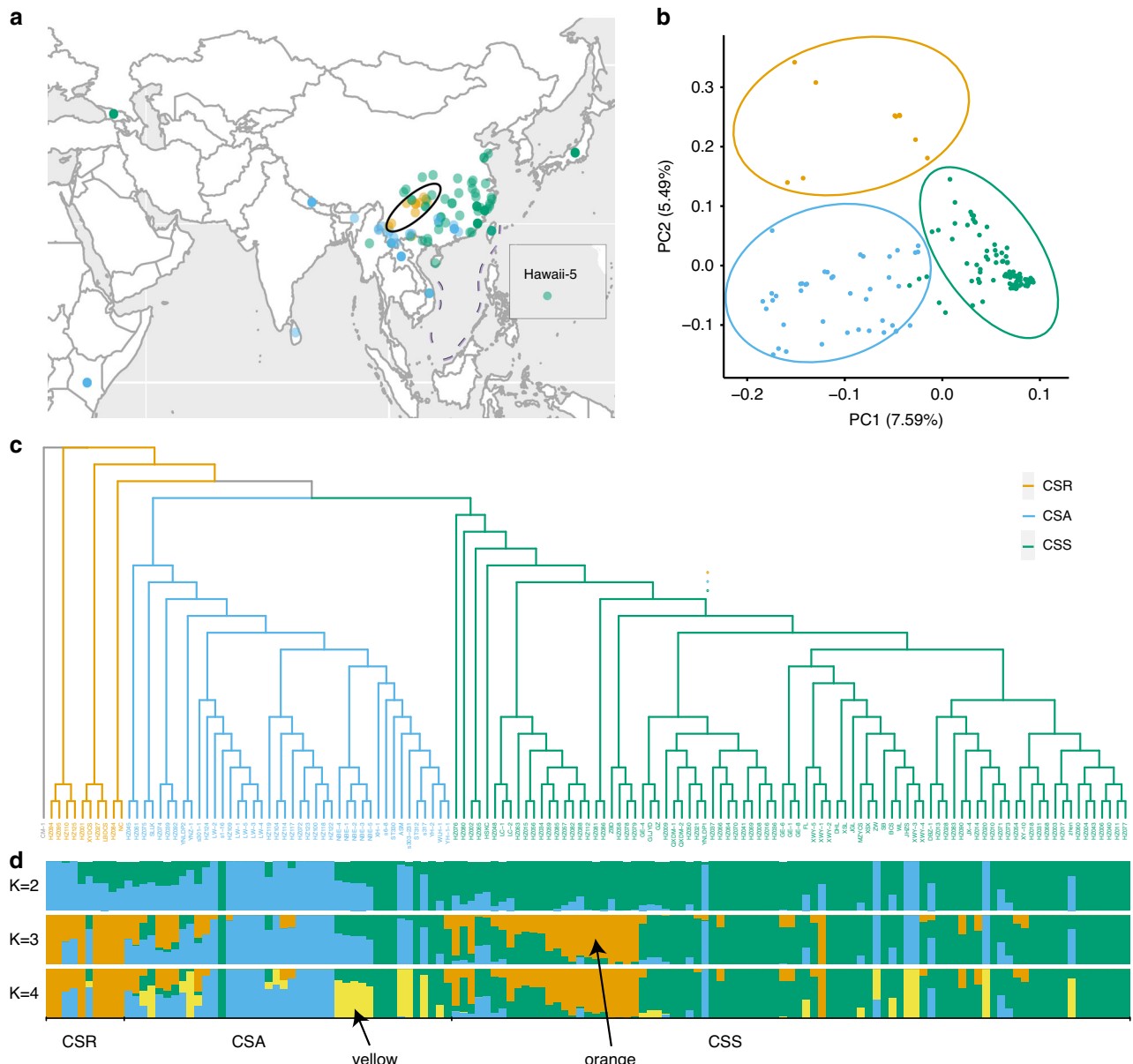

**Fig. 2 Distribution and evolution of tea. a** The distribution of tea accessions assessed in the present study. The teas within the black oval had the highest nucleotide polymorphism. **b** Principal component analysis of the tea populations. PC1 and PC2 split the tea populations into three clusters. The *Camellia sinensis* var. *sinensis* (CSS) samples were found to cluster more tightly than the *C. sinensis* var. *assamica* (CSA) samples. **c** A phylogenetic tree of tea. *Camellia sasanqua* Thunb. was used as the outgroup, and the tea samples closest to the outgroup were *C. sinensis*-related species (CSR). **d** Structure of the tea populations. Green, blue, and yellow represent the CSS, CSA, and CSR populations, respectively. Yellow and orange are marked with arrows. Source data are provided as a Source Data file.

annotations were complete, compared with the 1108 (80.58%) and 943 (68.58%) complete annotations obtained for the SCZ and YK10 genomes, respectively.

Using the genome annotation data, we determined the chromosomal locations of 26,561 (79.15%) annotated genes. Furthermore, we compared the protein sequences of the LJ43 genome with those of *Actinidia chinensis*, a species in order Ericales for which a high-quality reference genome sequence is available, and used MCScanX to detect synteny (Supplementary Fig. 8). The results revealed that the LJ43 genome comprises 690 collinear blocks containing 18,030 genes, whereas the SCZ genome has 111 collinear blocks containing 1487 genes and that of YK10 has 54 collinear blocks containing 393 genes. Furthermore, we found that the extent of genome synteny between LJ43 and

cocoa (*Theobroma cacao*) was comparable to that with *Actinidia chinensis* (Supplementary Note 3).

**Gene family evolution**. To gain insight into the evolution of the tea genome, we grouped orthologous genes using OrthoMCL and accordingly obtained 24,350 groups of orthologous gene families among nine genomes (Supplementary Note 3). In total, 1034 single-copy gene families were used to construct a phylogenetic tree for the tea genome, using *Amborella trichopoda* as an outgroup (Supplementary Fig. 9). Gene family evolution was analyzed using CAFE, which revealed that a total of 1936 and 1510 tea gene families have undergone expansion and contraction, respectively. Gene Ontology (GO), InterPro (IPR), and Kyoto Encyclopedia of Genes and Genomes

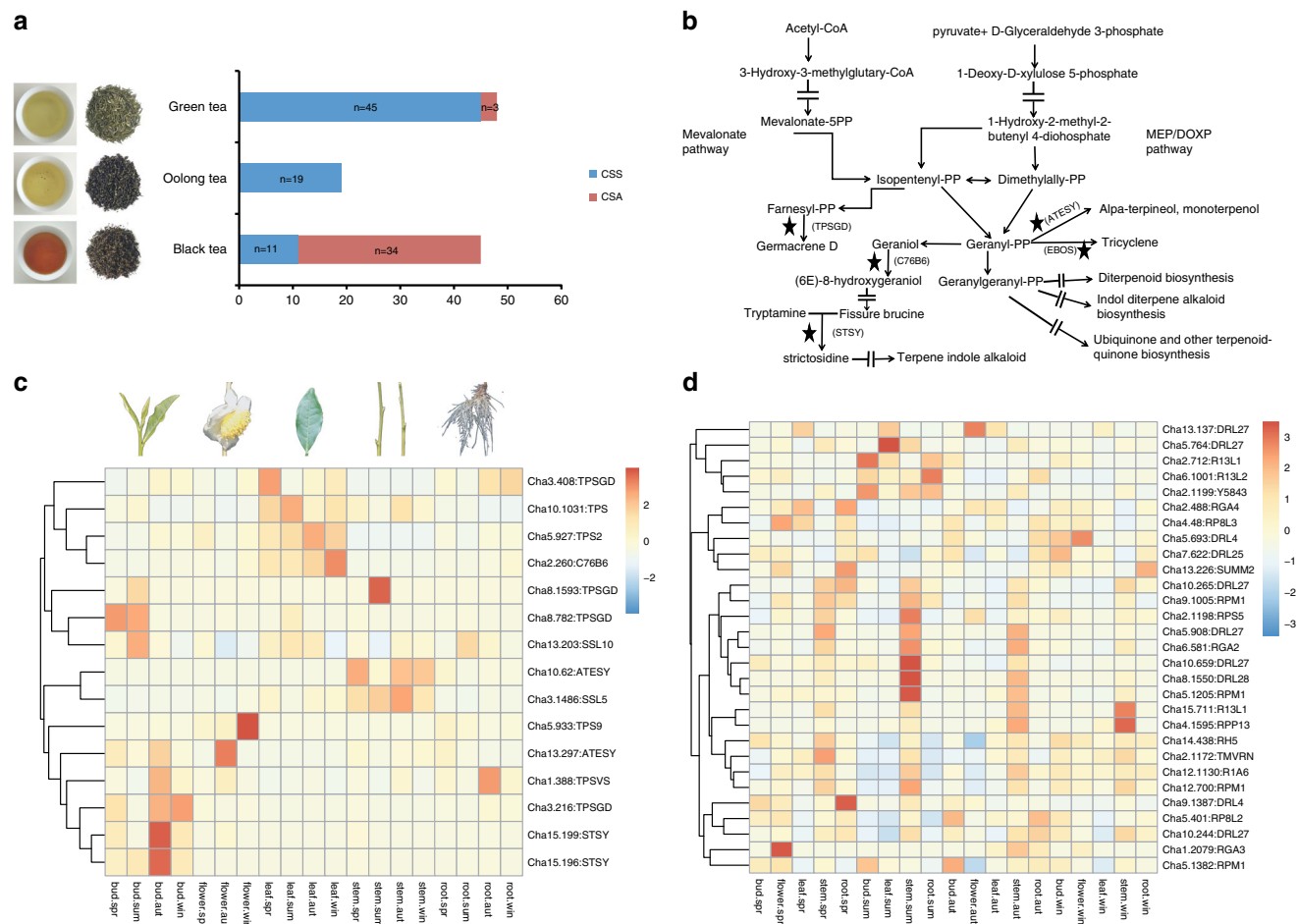

**Fig. 3 Sweep genesets in CSA and CSS show different directions of domestication. a** The tea types were used for SweepFinder2 analysis of CSS (*C. sinensis* var. *sinensis*) and CSA (*Camellia sinensis* var. *assamica*). **b** The pathway of terpene metabolism. The selective sweep genes are indicated by stars. The arrows bisected by equal symbols indicate hidden processes. **c** The expression of terpene-related genes in different tea tissues. **d** The expression of *NBS-ARC* genes in different tea tissues. Source data underlying Fig. 3c, d are provided as a Source Data file.

(KEGG) enrichment analyses of the expanded genes revealed the expansion of gene families involved in disease resistance, secondary metabolism, and growth and development (*P* value < 0.05, FDR < 0.05; Supplementary Tables 8–13; Supplementary Data 1). These families included UDP-glucuronosyl/UDP-glucosyltransferase (GO:0016758, *P* value < 2.20E-16, FDR < 2.40E-14), which catalyzes glucosyl transfer in flavanone metabolism and is related to catechin content; (−)-germacrene D synthase (K15803, *P* value = 8.01E-06, FDR = 0.91E-03), which catalyzes the conversion of farneyl-PP to germacrene D and is related to terpene metabolism; NB-ARC (GO:0043531, *P* value < 2.20E-16, FDR < 2.40E-14), Bet v I/Major latex protein (GO:0009607, *P* value = 4.49E-04, FDR = 8.64E-03), RPM1 (K13457, *P* value < 2.20E-16, FDR < 1.25E-13), and RPS2 (K13459, *P* value = 8.88E-08, FDR = 2.51E-05), which are related to disease resistance; and the S-locus glycoprotein domain (GO:0048544, *P* value < 2.20E-16, FDR < 2.40E-14), which is associated with self-incompatibility.

Furthermore, we used the "branch-site" models A and Test2 to identify the genes in the tea genome that have evolved under positive Darwinian selection using codeml in the PAML package (version 4.9d). A total of 1031 single-copy genes from the nine aforementioned genomes were scanned to identify genes under selection. After filtering (see the Methods section for details), we identified 74 genes that appeared to be under positive selection (FDR ≤ 0.05; Supplementary Data 2), some of which are involved in disease resistance, enhanced cold tolerance, and high-light

tolerance. In this regard, it has previously been reported that overexpression of cationic peroxidase 3 (OCP3)[27] (Cha14g001590) and Serpin-ZX[28] (Cha09g003010) is involved in disease resistance, whereas that of beta-glucosidase-like SFR2 (SFR2, Cha05g001710) is involved in freezing tolerance[29]. Other identified genes included one involved in the maintenance of photosystem II under high-light conditions (MPH1[30], ChaUn21494.1) and a photosystem II 22-kDa protein (PSBS, Cha09g008070) that protects plants against photooxidative damage.

**Whole-genome duplication and genomic divergence in tea**. To estimate the whole-genome duplication in tea, we selected a total of 3373, 3199, and 2992 gene families containing exactly two paralogous genes from the SCZ, LJ43, and YK10 genomes, respectively, to calculate the Ks values of the gene pairs. The results showed that the Ks peak of the three tea genomes was 0.3 (Supplementary Fig. 10), and the most recent duplication time was ~25 mya (Time = Ks/2μ, μ = 6.1E-9)[31], thereby indicating that these cultivars underwent the same genome duplication event. Syntenic genes between LJ43 and SCZ and between LJ43 and YK10 were identified to calculate the Ks values of the pairs, which indicated Ks peaks of ~0.003 (~0.25 mya) for the LJ43 and SCZ pairs and ~0.045 (~3.69 mya) for the LJ43 and YK10 pairs (Supplementary Fig. 11), thus indicating that the divergence times of LJ43 and SCZ were more recent than those of LJ43 and YK10.

**Population genetic analysis.** Tea leaves from different species or cultivars are often processed to produce different types of teas according to their processing suitability and local consumer preferences; e.g., CSA leaves are often processed to produce black tea, whereas CSS leaves are typically processed to produce green or oolong tea. To investigate the genetic basis of these differences, we examined the genomes of 139 tea accessions collected from around the world (Fig. 2a; Supplementary Data 3; Supplementary Note 4). The specimens were sequenced at an average depth of 13.67-fold per genome (Supplementary Data 3). Given that the LJ43 genome is well annotated and has a high level of continuity, we selected this genome as the reference genome. We accordingly achieved an average mapping rate of 99.07%, with minimum and maximum rates of 96.95% and 99.66%, respectively (Supplementary Data 3). After performing five filtering steps (described in the Methods section), we identified a total of 218.87 million SNPs among the tea populations, with a density of approximately 67 SNPs per kb (Fig. 1a; Supplementary Tables 14 and 15). We anticipate that this extensive whole-profile SNP dataset will be valuable for further tea genomics research and marker-assisted breeding.

To further investigate the phylogenetic relationships among these accessions, we constructed a maximum likelihood-based phylogenetic tree with SNPs filtered from the total SNP dataset (see the Methods section for details), using *Camellia sasanqua* as an outgroup (Fig. 2c). We found that all samples were clustered into one of three independent clades (Fig. 2c; Supplementary Data 4) corresponding to the CSR, CSS, and CSA populations, which is consistent with the morphology-based classical taxonomy of CSA and CSS.

Principal component analysis (PCA) was used to investigate the relationships and differentiation among populations and consistently revealed the presence of three clusters corresponding to CSA, CSS, and CSR (Fig. 2b). The first two principal components accounted for 13.08% of the total variance, with PC1 reflecting the variability of the CSA and CSS groups and PC2 differentiating CSR plants from CSA and CSS plants. We found that CSS showed better aggregation than CSA and CSR, whereas the juncture accessions of CSA and CSS were also close to CSR in the phylogenetic tree. At a K value of 3, CSA, CSS, and CSR could be readily distinguished (Fig. 2d; Supplementary Fig. 12; Supplementary Note 4), which is consistent with the PCA results (Fig. 2b). At a K value of 3 or 4, most new accessions collected from outside China appeared to have originated from CSA and CSS (yellow color, marked with an arrow in Fig. 2d), indicating their high diversity.

On the basis of the phylogenetic and population structure results (Fig. 2c; Supplementary Data 4–6), we further investigated individual- and population-level heterozygosity among the populations (Supplementary Data 3). We accordingly found the heterozygosity of CSR (6.37E-3) to be significantly higher than that of CSA (6.29E-3) and CSS (5.69E-3) (both $P$ values < 0.05; Supplementary Fig. 13). We also calculated linkage disequilibrium (LD) decay values based on the squared correlation coefficient ($r^2$) of pairwise SNPs in two groups, which revealed that for the CSA and CSS groups, the average $r^2$ among SNPs decayed to ~50% of its maximum value at ~41 and 59 kb, respectively. These values thus indicate that the tea genomes have relatively long LD distances and slow LD decay (Supplementary Fig. 14).

**Selective sweeps in the two major tea populations.** It is generally thought that the differences between CSS and CSA teas lie primarily in their flavor, leaf and tree types, cold tolerance, and processing suitability. Among the accessions assessed in the present study, the CSA population comprised three green tea accessions and 34 black tea accessions, whereas the CSS population contained 45 green tea accessions, 19 oolong tea accessions, and 11 black tea accessions (Fig. 3a). To determine the potential genetic bases of these differences, we used SweepFinder2 (version 1.0) to scan for selective sweep regions and selected regions with the top 1% of composite likelihood ratio (CLR) scores and the genes overlapping with the final sweep regions (≥300 bp). On the basis of this analysis, we identified a total of 1336 and 1028 genes bearing selection signatures in the CSA and CSS populations, respectively (Supplementary Data 7 and 8; Supplementary Fig. 15).

Using the data generated from GO analysis, we selected enriched genes ($P$ value < 0.05, FDR < 0.05) from the candidate selective sweep genes of the CSA and CSS populations (Supplementary Tables 16 and 17; Supplementary Fig. 16) and accordingly found that volatile terpene metabolism genes, such as cytochrome P450s (e.g., geraniol 8-hydroxylase) and terpene synthases, including alpha-terpineol synthase (*ATESY*), (−)-germacrene D synthase (*TPSGD*), and strictosidine synthase (*STSY*), were significantly selected in the CSS population but not the CSA population (Fig. 3b; Supplementary Tables 16 and 17). The functionalization of core terpene molecules requires cytochrome P450s[32], among which geraniol 8-hydroxylase catalyzes the conversion of geraniol (6E)-8-hydroxygeraniol (Fig. 3b), which may affect the accumulation level of geraniol. Alpha-terpineol, a monoterpene found in tea, is generated by the ATESY-mediated catalysis of geranyl-PP, whereas TPSGD catalyzes the conversion of farneyl-PP to the sesquiterpene germacrene D. Strictosidine is the precursor of terpenoid indole alkaloids, and STSY is a key enzyme in the synthesis of these alkaloids (Fig. 3b). Moreover, we found that 80% of the selected terpene-related genes showed relatively high expression in buds or leaves, whereas 33% of these genes showed significantly high expression in buds or leaves (Fig. 3c; Supplementary Table 18).

Compared with the CSA accessions, the CSS accessions were characterized by the selection of a larger number of *NBS-ARC* (nucleotide-binding site domain in apoptotic protease-activating factor-1, R proteins and *Caenorhabditis elegans* death-4 protein) genes, the *Arabidopsis* homologs of which, including *RPS3* (also known as *RPM1*)[33], *RPS5*[34], and *SUMM2*[35], have been shown to be involved in resistance to *Pseudomonas syringae* (*RPS*) (Supplementary Tables 16 and 17). The expression profiles of these genes revealed that 69% of the *NBS-ARC* genes subject to selection are highly expressed in spring, autumn, or winter, whereas 24% of these genes are significantly highly expressed in spring, autumn, or winter (Fig. 3d; Supplementary Table 19). However, among the 214 genes under selection in both the CSS and CSA populations, we were unable to detect enrichment of any genes related to flavor synthesis or abiotic and biotic stress resistance in the CSA population (Supplementary Data 7 and 8).

## Discussion

This study presents a chromosome-scale genome sequence of tea and resequencing data for 139 tea accessions collected from around the world. According to our analyses, these genomic resources will be valuable for future genomics research and molecular breeding of tea. The data reveal the genome-wide phylogeny of tea and the directions of divergent selection between the two main tea varieties, namely, CSS and CSA. Compared with CSA, in CSS, genes involved in flavor metabolism and cold tolerance have been subjected to stronger selection, which is consistent with the fact that tea accessions from eastern and northern China, such as green and oolong tea, have a distinct aroma and are cold tolerant. Our data also indicate that the CSR population

is an ancestor of CSS and CSA. However, although these findings represent an important step in unravelling details of the origin and domestication of CSS and CSA, it remains necessary to identify the closest ancestor of tea and to examine a larger number of CSR accessions in the future. Due to the limitations of sampling in India, we cannot rule out the possibility of other evolutionary scenarios, an evaluation of which will require a more comprehensive collection of samples. Although several studies related to tea genomics have recently been published[1,16,36,37], considering the complex genome content and valuable contribution to both agriculture and health of tea, a future pangenome analysis will be essential for the tea research community.

The first important step in a genome sequencing project is to obtain a high-quality reference genome and call an SNP set with high confidence from well-mapped resequencing data. In this regard, the inherent characteristics of the tea genome, notably its large size, high heterozygosity (Supplementary Table 20), and large number of repetitive sequences (Supplementary Tables 5 and 6), have hampered genome assembly. Although the genomes of the YK10 and SCZ tea cultivars were previously reported, they were characterized by low continuity compared with that of the major currently assembled genomes (Mb scale) at both the contig and scaffold levels. Moreover, the associated BUSCO scores indicated that only ~80% of the predicted genes could be identified in these genomes. Taking advantage of recent advances in sequencing and assembly technologies, we were able to sequence the genome of the LJ43 tea cultivar at the chromosome scale, generating an assembly characterized by a scaffold N50 value of 144 Mb, 88.36% gene completeness, and a base accuracy of 99.999%. Nevertheless, given the complex nature of the tea genome, further improvement in genome annotation will be desirable in the future. Combined with other analyses, our results showed that the quality of the LJ43 genome is higher than that of previously published tea genomes[1,16]. Furthermore, our whole-genome sequencing of 139 worldwide tea accessions generated 6272.74 Gb of short reads and 218.87 million high-confidence SNPs, and collectively, the datasets obtained in the present study provide the richest genomic resource for tea researchers compiled to date.

*Camellia* is one of the most taxonomically and phylogenetically challenging plant taxa[12], as there are many disparities between traditional morphology- and whole-genome sequence-based analyses. Gene flow is notably widespread among tea accessions (Supplementary Note 4; Supplementary Table 21; Supplementary Data 9 and 10; Supplementary Fig. 16), which presents challenges for determining the origin and evolution of tea. For example, *C. taliensis* (HZ122, HZ114) and *C. gymnogyna* (HZ104) were previously assigned to the CSA population. Bitter tea, a hybrid progeny of CSS and CSA teas[38], is a transitional type of large-leaved tea with a growth habit ranging from tree-like to shrub-like and is mainly distributed in areas with mixed growth of CSS and CSA. In our phylogenetic tree, bitter teas (HZ039, HZ092, HZ080, and HSKC) were closely clustered with transitive teas in CSS and CSA, supporting bitter tea as a hybrid progeny of CSS and CSA. It is expected that further worldwide sampling and more comprehensive data analysis will help resolve the current debates concerning tea taxonomy.

Unlike annual crops or perennial self-compatible crops, tea has not experienced severe domestication bottlenecks between wild progenitors and cultivated varieties[39]. Our results showing no signs of a clear bottleneck in the last 20,000 years (Supplementary Note 4; Supplementary Figs. 17 and 18) are conceivably attributable to the widespread gene flow between different teas, which may have masked recent bottlenecks or may indicate that tea plants have not experienced strong domestication-related selection, and is also indicative of the complex history of tea evolution and

domestication. During the expansion and domestication of tea, cultivated teas have been crossed with wild relatives, which contributed to the current genetic complexity of tea populations. This interbreeding is reflected by our observation that many cultivars and wild resources clustered together in the phylogenetic tree, with ancestral wild relatives appearing in the CSS cluster when a K value of 3 was used in the structural analysis (Supplementary Note 4; Supplementary Table 21; Supplementary Data 3, 9, and 10; Supplementary Figs. 12 and 16). Although China has the longest history of tea cultivation and the oldest written literature[4,40,41] to support the hypothesis that tea plants originated in this country, there is still a lack of consensus regarding the events associated with the domestication of tea. In this regard, Meegahakumbura et al. suggested that the origins of CSS and CSA in China and CSA in India can probably be traced to three independent domestication events in three separate regions across the two countries[42,43]. However, this statement should be interpreted with caution given the lack of available progenitor species of both CSS and CSA. Our data indicated that the CSR population is an ancestor of CSS and CSA. However, there remains a necessity to identify the closest ancestor of tea and to examine a larger number of CSR accessions in the future.

In the present study, we identified two interesting selection signatures in the CSS population, one of which was associated with genes involved in the terpene synthesis pathway. Terpene volatiles play essential roles in defining the characteristic aroma of tea, and the compositions and concentrations of these volatiles are controlled at the genetic level[44]. Different species or varieties of tea plants are characterized by differences in terpene profiles. Takeo et al.[45–47] found that the contents and ratios of linalool and its oxides were high in CSA, whereas the contents and ratios of geraniol and nerolidol were high in CSS. The main terpenoids determining the aroma of black tea are linalool and its oxides, whereas geraniol and nerolidol contribute to the aroma of green tea and oolong tea[48]. These distinctions are consistent with the findings of our population selection analysis, which revealed that the terpene metabolism genes geraniol 8-hydroxylase, *ATESY*, *TPSGD*, and *STSY* were significantly selected. In addition, our KEGG enrichment analysis of expanded gene families revealed that *TPSGD* is expanded in the LJ43 cultivar at the genomic level. Moreover, the flavor of different tea types has been influenced to a certain extent by consumer predilection and culture. On the basis of the processing suitability of CSA and CSS and the population selection analysis of the two populations, we can conclude that terpenoid metabolism is more closely related to the aroma of green and oolong tea than it is to that of black tea.

The second selection signature of interest identified in the present study relates to *NB-ARC* genes in the CSS population. Most of these genes are associated with resistance to ice nucleation-active (INA) bacteria. In *Arabidopsis*, *RPS3*/*RPM*[33], *RPS5*[34], and *SUMM2*[35] have been shown to confer resistance to *Pseudomonas syringae*, which is one of the most well-studied plant pathogens and can infect almost all economically important crop species. In addition, *P. syringae* is a prominent INA bacterium and has been proposed to be an essential factor contributing to frost injury in agricultural crops[49]. Mutants characterized by alterations in the aforementioned genes have also been found to show sensitivity to chilling temperature compared with corresponding wild-type plants[33–35]. Similarly, in wild potato (*Solanum bulbocastanum*), the *RGA2*[50] and *R1A6* genes are involved in resistance to *Phytophthora infestans*, a further factor related to INA bacteria. Moreover, significant differences have been detected in the expression of *RPS3* and *SUMM2* in cold-resistant and cold-susceptible cultivars[51]. Taken together, the results of these studies suggest that *NB-ARC* genes might play an important role in endowing CSS cultivars with cold

tolerance. Tea grown along the Yangtze River Basin and in eastern China is typically subjected to low temperatures in early spring and winter, and most CSA cultivars, which are characterized by large leaves, cannot survive in these areas. Some CSS cultivars adapted to cold environments survived during expansion and domestication in eastern and northern China, and after the separation of CSS and CSA, the direction of domestication of these two varieties is assumed to have diverged. With increasing tea consumption, humans began to select tea plants, and during domestication, selection for flavor and cold tolerance was stronger in CSS than in CSA. This is also reflected at the genomic level, as illustrated by the KEGG enrichment of expanded gene families, in which the disease resistance proteins RPS2 and RPS3 were found to be expanded in LJ43.

Although we found that 214 genes had undergone selection in both the CSS and CSA populations in the present study, we were unable to detect enrichment of any of the genes associated with flavor and resistance in the CSA population (Supplementary Table 16). Thus, these observations indicate that selection for INA bacterial resistance and flavor during domestication has been stronger in CSS than in CSA.

## Methods

**Materials and sequencing**. We collected samples of 139 tea accessions from around the world (detailed information is presented in Supplementary Data 3). Among samples, 93 were collected from China, with the remaining 46 samples being obtained from the other main tea-producing countries. For the purpose of analyses, we selected *Camellia sasanqua* Thunb. as outgroup. DNA was extracted from the leaf tissues of all samples using the CTAB method[52]. Libraries for Illumina TruSeq, 10× Genomics, and PacBio analyses were prepared according to the respective manufacturer's instructions. Details of the sequencing procedure are presented in Supplementary Note 1.

**Genome assembly and annotation**. Details regarding genome size and genome assembly are presented in Supplementary Note 1. Assembly of the LJ43 genome was performed based on the Hi–C-Pro pipeline and full PacBio reads using WTDBG (version 1.2.8). The final Hi–C-assisted genome assembly was commissioned by Annoroad Gene Technology. Tigmint (version 1.1.2)[53] was used to detect errors using linked reads from 10× Genomics Chromium. The reads were initially aligned to the Hi–C scaffolds, and the extents of the large DNA molecules were inferred from alignments of the reads. For larger-scale gaps, we mapped optical maps from BioNano Genomics to the Hi–C scaffolds using the BioNano Solve 3.3 analysis pipeline. A high-density genetic linkage map[20] was used for genomic synteny analysis. The markers were initially aligned to the Hi–C scaffolds using bwa mem (version 0.7.15), and correctly mapped alignments with a mapping quality >1 were extracted (3483). Dot plots were generated, and correlations with the extracted alignments were calculated using R (version 3.4). Repeat sequences were identified using de novo and homology-based methods. Augustus and GlimmerHMM were used for ab initio gene prediction with parameters trained using unigenes. For homology-based predictions, we used the homologous proteins proposed for the genomes of *Arabidopsis thaliana*, *Oryza sativa* subsp. *japonica*, *Coffea canephora*, *Theobroma cacao*, and *Vitis vinifera*. RNA was extracted from five tissue types (bud, leaf, flower, stem, and root) at four time points corresponding to the seasons (with the exception of flowers during summer). Three biological replicates were prepared for each sample (Supplementary Table 7), and the transcript reads were assembled using Cufflinks (version 2.2.1). All of the predicted gene structures were integrated using EVidenceModeler (version 1.1.1). Protein-coding genes with a coding sequence length shorter than 300 nt and with stop codons were filtered (with the exception of stop codons at the end of a sequence). We then mapped RNA-seq reads to the predicted coding regions using SOAP2 and selected the predicted gene regions based on RNA-seq data (regions with >50% coverage). The methods used for gene and functional annotation are described in detail in Supplementary Note 2. The sequences of LJ43 and *Actinidia chinensis*[21] proteins were analyzed using blastp with the parameters -evalue 1e-5 -num_alignments 5. Thereafter, syntenic blocks were identified using MCScanX with the parameters –e 1e-20. SCZ and YK10 were analyzed using the same pipeline and parameters. We also analyzed the genome synteny between *Theobroma cacao*[54] and LJ43, SCZ, and YK10 (Supplementary Note 3).

**Analysis of positive Darwinian selection**. A species tree was constructed as described in Supplementary Note 3, without SCZ and YK10. We identified 1031 single-copy gene families. The protein sequences of single-copy genes were aligned using ClustalW2[55], and then the ClustalW2 data were transformed to nuclear format according to the alignment protein sequences using an in-house Perl script. Gblocks[56] was used to cleave the nuclear alignment sequences based on the $t = c$

parameter. "Branch-site" models A and Test2 were selected to assess positive selection using codeml of the PAML package. The significant sites were dropped if the 5-bp sequences around the site sequences were cut by Gblocks. A false discovery rate (FDR) value of ≤0.05 was used to filter the results.

**SNP calling and filtering**. Quality-controlled reads were mapped to the unmasked tea genome using bwa (version 0.7.15)[57] with the default parameters. SAMtools (version 1.4)[58] was used for sorting, and Picard (v.2.17.0) was used to remove duplicates. The HaplotypeCaller of GATK (version 3.8.0)[58] was used to construct general variant calling files for the tea group (139 accessions) and outgroup (*C. sasanqua*, CM-1) by invoking -ERC:GVCF. gVCF files in the tea group were combined using GenotypeGVCFs in GATK to form a single-variant calling file, whereas the gVCF file for the outgroup was called using the option '–allSites' to include all sites. The final single-variant calling file was merged using BCFtools (version 1.6), with only the consistent positions retained in both groups. To obtain high-quality SNPs, we initially used the GATK hard filter to filter the merged VCF data with the options (QD ≥ 2.0 && FS ≤ 60.0 && MQ ≥ 40.0 && MQRankSum ≥ −12.5 && ReadPosRankSum ≥ −8.0). Thereafter, we performed strict filtering of the SNP calls based on the following criteria: (1) sites were located at a distance of least 5 bp from a predicted insertion/deletion; (2) the consensus quality was ≥40; (3) the sites were not trialleic and did not contain InDels; (4) the depth ranged from 2.5 to 97.5% in the depth quartile; and (5) SNPs had minor allele frequencies (MAFs) ≥ 0.01.

**Population genetic analyses**. We selected high-quality SNPs with a maximum of 20% missing data, and to eliminate the potential effects of physical linkage among variants, the sites were thinned such that no two sites were within the same 2000-bp region. Phylogenetic analysis was conducted with the final SNP set using IQ-TREE (version 1.6.9)[59–61]. A maximum likelihood (ML)-based phylogenetic tree was constructed using the GTR + F + R5 model, with 1000 rapid bootstrap replicates conducted to determine branch confidence values. The best-fitting model was estimated using ModelFinder implemented in IQ-TREE after evaluating 286 DNA models. GTR + F + R5 was selected based on the Bayesian information criterion. The ML phylogenetic tree was constructed based on intergene region SNPs using the final SNP set and 4DTV SNPs. Principal component analysis (PCA) of the final SNP set was performed using PLINK (version 1.90), with the principal components plotted against one another using R 3.4 to visualize patterns of genetic variation. We also used the final SNP set for population structure analysis using ADMIXTURE (version 1.3)[62], which was run with K values (the number of assumed ancestral components) ranging from 1 to 10.

Population heterozygosity at a given locus was computed as the fraction of heterozygous individuals among all individuals in a given population. The average heterozygosity was then calculated for each 40-kb sliding window, with a step size of 20 kb. Individual heterozygosity was computed as the fraction of loci that were heterozygous in an individual. Average heterozygosity was also calculated using the same method. Windows with an average depth <1 were filtered out.

To eliminate the influence of differences in sample number, eight samples of the CSR/CSA/CSS populations were randomly selected to calculate nucleotide diversity. To reduce the sampling error, we performed 20 repeat calculations for each population using VCFtools (version 0.1.16) with a window size of 50 kb and a step size of 10 kb. The data for each population are presented as boxplots created using R.

**Selective sweep analysis**. TreeTime 0.5.3[63] was used to infer the ancestral state based on ML using the generated evolutionary tree. Sites lacking a reconstructed ancestral state in a population were folded in the SweepFinder2 analysis. We excluded sites that were neither polymorphic nor substitutions, as recommended by the SweepFinder2 manual[64]. To reduce the likelihood of false positives, the chromosome-wide frequency spectrum was calculated as the background for each chromosome and population. SweepFinder2 was run with a grid size of 100. The CLR scores from the SweepFinder2 results were extracted and merged into sweep regions when the neighboring score(s) exceeded a certain threshold, which was set as the top 1% of CLR scores. To obtain regions with greater continuity, we merged regions into a single region with a certain size threshold between regions, with the threshold being set to 50% of the size in the adjacent sweep regions. The final score for each sweep region was the sum of the CLR scores of the sites in the sweep region. The final sweep regions were filtered based on a minimum size of 300 bp. Genes overlapping within the sweep regions were extracted as candidate selective sweep genes. The GO-enriched (P value < 0.05, FDR < 0.05) candidate selective sweep genes were chosen, and *Fst*, $\theta_\pi$ and Tajima's D values were calculated using VCFtools with a window size of 50,000 bp and a step size of 10,000 bp.

**Gene expression**. Transcript-level expression was calculated using HISAT2, StringTie, and Ballgown with the default parameters[65]. The genes identified among the selection results were selected for expression analysis, and an expression heatmap was plotted using the heatmap package in R 3.4. The average expression of selected genes shown in Fig. 3d was calculated according to season, whereas the average expression of selected genes shown in Fig. 3c was calculated according to

tissue. Student's *t*-test was used to identify the significantly differentially expressed genes (P value < 0.05).

**Reporting summary**. Further information on research design is available in the Nature Research Reporting Summary linked to this article.

## Data availability

The RNA-seq, 10× Genomics, Hi–C, Illumina short reads, and PacBio raw data for the 'Longjing 43' cultivar of *Camellia sinensis* var. *sinensis* have been deposited in the European Bioinformatics Institute with the accession code PRJEB39502. The raw resequencing data have been deposited in the National Center for Biotechnology Information (NCBI) Sequence Read Archive database with the accession codes PRJNA646044. All raw sequence data are also available in the Genome Sequence Archive[66] in the BIG Data Center[67], Beijing Institute of Genomics (BIG), Chinese Academy of Sciences, under accession number PRJCA001158. The assembly and annotation of the 'Longjing43' genome are available in BIG database [https://bigd.big.ac.cn/search/?dbId=gwh&q=GWHACFB00000000]. Source data are provided with this paper.

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

## Acknowledgements
This work was funded by the Agricultural Science and Technology Innovation Program of the Chinese Academy of Agricultural Sciences (CAAS-ASTIP-2017-TRICAAS and CAAS-ASTIP-2017-AGISCAAS), the Agricultural Science and Technology Innovation Program Cooperation and Innovation Mission (CAAS-XTCX2016), the Major Science and Technology Special Project of Variety Breeding of Zhejiang Province (2016C02053), Shenzhen Science and Technology Research Funding (JSGG20160429104101251), the National Youth Talent Support Program and the Program for the Innovative Research Team of Yunnan Province. We thank Hualin Huang (Tea Research Institute of Guangdong Agricultural Academy of Sciences), Haitao Zheng (Rizhao Tea Research Institute), and Lizhe Lv (Xinyang Tea Research Institute) for supplying tea plant samples. We thank Xiujuan Shao for analyzing the gene annotations of LJ43. We thank Assistant Professor Supriyo Basak at the Kunming Institute of Botany, CAS, and Banasthali Vidyapith for help in assessing genome size with flow cytometry.

## Author contributions
X.W., Y.C., G.W., L.C., J.R., and Y.Y. designed the experiments and managed the project. X.W., F.H., Y.C., C.M., L.Y.W., X.H., and A.L. wrote the manuscript with input from all authors. X.W., F.H., Y.C., C.M., X.H., A.L., H.C., J.J., L.W., K.W., X.B.W., C.A., Z.W., S.Z., P.C., Y.L., B.L., G.W., L.C., J.R., and Y.Y. collected the samples, extracted genetic material, analyzed the data, and performed the experiments. X.W., Y.C., C.M., X.H., and S.Z. performed the experiments and genomic and RNA sequencing. J.R. performed the genome assembly analyses. H.F. and X.B.W. performed the gene annotation analyses. H.F., X.H., A.L., and C.A. performed transcriptomic analyses. X.W., H.F., A.L., and G.W. performed population analyses. X.W., Y.C., P.C., L.C., G.W., J.R., and Y.Y. revised the manuscript.

## Competing interests
The authors declare no competing interests.
