## [Peer Review File · Nature Communications]

Reviewer #1 [Editor: this reviewer signed off when this manuscript was under consideration by another Nature family journal.]

Reviewer #2 (Remarks to the Author):

The use of English in the revised manuscript is much improved and several of my previous points have been addressed, however some minor issues remain.

Q25

The authors have compared mapping rates with different software as requested. I agree that parameters and filtering are important, however the results presented clearly show that there is a major difference in mapping rate between BWA, which tends to overmap, and Bowtie (default) which is more conservative in how it places reads. From my experience, Bowtie results more accurately reflect true read mapping in plant genomes, which are often polyploid and highly complex. The authors should understand for read mapping, larger numbers are not always better and should account for this in their analysis.

Q27

The authors have performed several tests to support the view that tea has very large genes, however I remain unconvinced. For example, the long gene sample has 42% of exon junctions confirmed while the short gene list has 50% of exons conserved. Given that these are highly expressed genes, I would expect more exons to be conserved in both samples. Having fewer exons conserved in long genes does support the view that many of these are concatenated. The comparison of 16 genes does not make sense the way it is written and it is unclear exactly what was done. The comparison of genes with other species is flawed as the genes are filtered for coverage (concatenated genes would naturally only have partial coverage). Regarding so many genes not being on pseudomolecules, the authors address this by comparing repeat content with that of maize. Presumably comparing gene content on maize pseudomolecules and additional contigs would be more relevant. Continuing to claim that the assembly is fragmented due to polyploidy and complexity without comparison of fragmentation with other complex genomes such as wheat fails to support the claim.

Reviewer #3 (Remarks to the Author):

I have read over comments from the authors related to the previous review. Thank you for the revised version of the manuscript. In this limited time I have been able to read things over I think most of my concerns have been addressed. One thing that should be discussed however, is the recent publication of another tea genome with 81 diverse accessions resequenced. I realized this is a recent publication while this manuscript has been in review, but it must be addressed in this publication. Specifically, novelty in the current manuscript not presented and how the genomes now compare.

Xia et al 2020 *Molecular Plant*: The reference genome of tea plant and resequencing of 81 diverse accessions provide insights into genome evolution and adaptation of tea plants

<https://www.cell.com/action/showPdf?pii=S1674-2052%2820%2930134-9>

Reviewer #4 (Remarks to the Author):

Tea plant (*Camelia sinensis*) has a rich, long and complex history that generated the extant populations of the major producing countries such as China and India, However, little is known on about the wild ancestor of this species (M. K. Meegahakumbura et al. 2016; Muditha K. Meegahakumbura et al. 2017; Yao et al. 2012; Kingdom-Ward 1950). Despite its diploid genome efforts to assembly its genome have been overwhelmed by the complexity of the genome adding the factor that it is self-incompatible making it highly heterozygous factor that increase the difficulty of the assembly (Wei et al. 2018). To this date, tea still a interesting model to study selection dynamics that lead to the local adaptation of the different populations as result of this wide dispersion.

In this manuscript, Xinchao Wang and co-authors achieved an improvement of the genomic reference when compared to the previous reference published in PNAS by Wei et al 2018, by using long read sequencing and the physical interaction provided by Hi-C data. Increasing the N50 size of the scaffolds from less than 1.4 Mb to 143.85 Mb, increasing the amount of core genic annotations by 7.8%, and anchoring around 2.3 Gbp out of 3.26 Gbp to chromosomal level. Also, they attempted to disentangle the domestication history of the crop by sequencing of 134 tea accessions that were grouped in 3 different populations (CSS, *C. sinensis* var. *sinensis*; CSA, *C. sinensis* var. *assamica* and CSR, *C. sinensis*-related species). They presented a evidence showing in the form of a PCA that the first two principal components explained 13.08% of the total variance observed between all the sampled individuals. Maximum Likelihood phylogenetic tree that suggested the group CSR as ancestral to the rest of the groups (CSS and CSA).

This manuscript provides a noticeable improvement of *Camelia sinensis* genome resources and also gives a complete insight of the evolution of the Chinese tea accessions and its dispersal across many countries.

Even that the analysis conducted in your manuscript were conducted with high quality and standards, the discussion section can benefit from the argumentation provided in this review.

for the discussion section I suggest you to consider the limitations of the sampling in regard to the India origin and the exploit the opportunities that the sampling has in terms of the expansion of the origin of two main populations of China tea and its dispersal across the world.

Major comments

Main text line 370 to 372. the argument claiming that the breeding of the plants has largely been determined by the environment rather than human behavior is attributed to a lack of severe domestication is not supported by clear evidence. the Supplementary Figs 20 and 21 show changes in population size through out time for the CSA and CSS clusters, however this is not an evidence that the environment created the given tendencies. in contrast Your F4 analysis suggest a strong gene flow between the accessions and is a evidence that the genetic diversity reduced by any bottleneck may be recovered as result of recombination masking any previous bottleneck.

To improve the discussion and overall interpretation of the Supplementary Fig. 20 I would include the CSR cluster into the Ne analysis to see the moment in which the populations from CSA and CSS change in size respect CSR. If the change of CSR respects the other groups occurred at a point where CSA and CSS where very similar that may be another evidence to suggest the origin of CSA and CSS happened at a similar time point. Also please make clear the X axis of that figure to clearly see the age before present of the time points; the given exponential notation complicates the interpretation.

Main test line 380 to 389. in these lines you describe that the evidence that CSR is ancestral to CSS and CSA make doubtful the observation of a probably independent domestication event in China and India provided by Meegahakumbura et al. 2016. I do not agree with your claim, as not a single accession from India was included in your analysis. Therefore, the possibility of an independent event still valid. What is very interesting about having CSR as ancestor of CSS and

CSA is that it may be an evidence of a unique origin of the CSR and CSA clusters that may be dispersed after this event as far as Africa, Georgia or Hawaii.

Supplementary materials 356 to 364. please include the error values for K clustering in the admixture analysis to confirm that choosing a K=3 is the most likely scenario of clustering and that there are not other more likely clusters.

REVIEWER COMMENTS

Reviewer #1 [Editor: this reviewer signed off when this manuscript was under consideration by another Nature family journal.]

Reviewer #2 (Remarks to the Author):

The use of English in the revised manuscript is much improved and several of my previous points have been addressed, however some minor issues remain.

Q1: The authors have compared mapping rates with different software as requested. I agree that parameters and filtering are important, however the results presented clearly show that there is a major difference in mapping rate between BWA, which tends to overmap, and Bowtie (default) which is more conservative in how it places reads. From my experience, Bowtie results more accurately reflect true read mapping in plant genomes, which are often polyploid and highly complex. The authors should understand for read mapping, larger numbers are not always better and should account for this in their analysis.

Author response: We are very grateful for sharing your valuable experience about the mapping tool. Before the starting of our project, we reviewed the plant resequencing articles published in high-level journals in recent years, and found that BWA was widely used, for examples, for the whole genome resequencing of apple (Duan et al. 2017), cotton (Du et al. 2018), upland cotton (Ma et al. 2018), pigeonpea (Varshney et al. 2017), silver birch (Salojarvi et al. 2017), and rice (Wang et al. 2018). Thus BWA was selected for our analysis. Bowtie was also an excellent mapping tool for NGS and had been used by many papers published. We already realized high mapping ratio is not necessarily good, so we had applied strict filtering rules to ensure the accuracy of SNP (manuscript lines 483-498). We will use Bowtie for our future whole genome sequencing projects according to your valuable suggestion.

Q2: The authors have performed several tests to support the view that tea has very large genes, however I remain unconvinced. For example, the long gene sample has 42% of exon junctions confirmed while the short gene list has 50% of exons conserved. Given that these are highly expressed genes, I would expect more exons to be conserved in both samples. Having fewer exons conserved in long genes does support the view that many of these are concatenated. The comparison of 16 genes does not make sense the way it is written and it is unclear exactly what was done. The comparison of genes with other species is flawed as the genes are filtered for coverage (concatenated genes would naturally only have partial coverage).

Regarding so many genes not being on pseudomolecules, the authors address this by comparing repeat content with that of maize. Presumably comparing gene content on maize pseudomolecules and additional contigs would be more relevant. Continuing to claim that the assembly is fragmented due to polyploidy and complexity without comparison of fragmentation with other complex genomes such as wheat fails to support the claim.

Author response: This question was actually made of 4 sub-questions, and we will response point to point to make it clearer.

Q2-1: The authors have performed several tests to support the view that tea has very large genes, however I remain unconvinced. For example, the long gene sample has 42% of exon junctions confirmed while the short gene list has 50% of exons conserved. Given that these are highly expressed genes, I would expect more exons to be conserved in both samples. Having fewer exons conserved in long genes does support the view that many of these are concatenated.

Author response: Thank you for your high accurate intuitive feeling to help us debug our analysis pipeline. Actually, when we get the value of exon junctions supported by RNA-seq, we also have the feeling that there should have something wrong with it, however, since most of the result for this analysis was as expected, we did not double-check the whole process. According to your valuable suggestion, we checked the analysis of exon junctions and found that we input the process file instead of our final file for genome annotation, in one step of our exon junction analysis. We apologize for this low-level mistake, and the corrected analysis is described here. We also double checked all of our analysis to ensure that there was no other mistake.

Our logic was that if the long gene among our gene annotation was reliable and not concatenation arose from error during genome assemble and annotation, the exon-exon conjunction in the longest and shortest gene set should have similar ratio supported with RNA-seq. Thus, we selected genes with highly expressed (FPKM values ≥ 20) in any tissue or period with more than two exons (14,698 genes). Among them, the top 1,000 longest genes contain 11,446 exon-exon conjunctions, and 9,289 (81.15%) were supported by RNA-seq reads. The 1,000 shortest genes contained 1,270 exon-exon conjunctions, and 932 (73.39%) were supported by transcription reads (in our previous analysis, we input the wrong file in this step). The exon-exon conjunctions in longer genes occupied higher percentage with RNA-seq support than shorter ones, this maybe the intrinsic character of transcriptome assembly with Illumina RNA-seq data (Adiconis et al. 2013; Nam et al. 2002), where the coverage at both of ends of the mRNA was lower than the gene body region (Fig 1 and 2). Thus the shorter gene will have higher percentage of exon-exon junction located at the gene's ends than the longer one.

We also selected genes with higher expressed (FPKM ≥ 100) in any tissue or period with more than two exons (4,060 genes). Among them, the top 1,000 longest genes contain 7204 exon-exon conjunctions, and 5925 (82.25%) were supported by RNA-seq reads. The 1,000 shortest genes contained 1,738 exon-exon conjunctions, and 1,434 (82.50%) were supported by transcription reads.

For genome *de novo* annotation, usually the gene model obtained with the support of RNA-seq data was regarded as most credible. The average genes length of transcriptome assembly was 11,417.3 bp, which was similar to final genes average

length (10,815.1 bp). Moreover, the average CDS length of LJ43 (1,205 bp) was similar to SCZ (1,345 bp) and YK10 (1,131 bp), if LJ43 contained many concatenated genes, the average CDS length would be significantly longer than SCZ and YK10.

Fig 1 The average relative coverage is shown at each relative position along the transcripts' length(Adiconis et al. 2013)

Fig 2 The average RNA-seq depth of relation position in genes of LJ43. X axis was the relation position in gene.

Q2-2: The comparison of 16 genes does not make sense the way it is written and it is unclear exactly what was done.

Author response: The comparison of the 16 genes as well as the *DAM2* was used to support the LJ43 genome annotation was reliable and better than SCZ and YK10, we agreed that this was not a valid proof and just want to use it as an extra proof. We selected these genes because this list was all of the current available gene with wet lab validation, most of these gene were validated by EST or RACE with reliable gene structure.

Q2-3: The comparison of genes with other species is flawed as the genes are filtered for coverage (concatenated genes would naturally only have partial coverage).

Author response: Thank you for pointing this out. We did not use coverage to filter the potential concatenated genes. We apologize for not supplying detailed information. We test the exon-exon junctions of suspected concatenated genes, if all exon-exon junctions were supported by RNA-seq reads, the suspected concatenated genes were excluded.

Q2-4: Regarding so many genes not being on pseudomolecules, the authors address this by comparing repeat content with that of maize. Presumably comparing gene content on maize pseudomolecules and additional contigs would be more relevant. Continuing to claim that the assembly is fragmented due to polyploidy and complexity without comparison of fragmentation with other complex genomes such as wheat fails to support the claim.

Author response: Thank you for pointing this out. For the question about anchor rate, in the beginning of using Hi-C to improve the genome assembly, high anchor ratio (>92%, Supplementary material lines 133-163, Supplementary Fig 3 and 4) could be obtained, but after visual inspection, we found that there were many mistakes in the resulted scaffolds (Supplementary Fig 3 and 4), which indicated that the high anchor ratio did not mean high quality and accuracy, so in the balance between the accuracy and the high contig anchor rate (longer Scaffold N50) of genome assembly, we preferred its accuracy for it will be more critical for further application in tea genomics and breeding research.

We agreed that the comparison of the assembly result between the tea and a more complex genome like wheat could support our opinion. However, the genome of wheat is larger than tea's, but the heterozygosity of wheat genome (AABBDD) is 0.0014 (Hussain et al. 2018), which was lower than LJ43 (0.0061). More importantly, the genome assembly strategies of wheat and tea were different, we listed the assembly strategies of wheat (Table 1). Thus, it was not suitable to compare the assembly result. The high heterozygosity and the exist of long repeat sequence of tea limited the continuity of genome assembly, the further improvement of the continuity of tea genome must rely on the development of sequencing technology in the future.

Table 1 The genome assembly strategies of wheat.

Wheat	Assembly strategies	Reference
Triticum urartu	BAC-by-BAC sequencing, Pacific Biosciences SMRT, BioNano genome map, 10× Genomics linked reads	(Ling et al. 2018)
Triticum Urartu	Illumina next-generation sequencing, BAC, EST sequences, and validation by PCR amplification	(Ling et al. 2013)
Chinese Spring	Flow-cytometric sorting, Illumina next-generation sequencing	(Mayer et al. 2014)
Aegilops tauschii	(Illumina next-generation sequencing	(Jia et al. 2013)

Durum wheat genetic map, Illumina HiSeq2500 platform, HiSeqX (Maccaferri et al. instrument, and chromosomes conformation capture 2019) sequencing (Hi-C)

Reviewer #3 (Remarks to the Author):

Q3: I have read over comments from the authors related to the previous review. Thank you for the revised version of the manuscript. In this limited time I have been able to read things over I think most of my concerns have been addressed. One thing that should be discussed however, is the recent publication of another tea genome with 81 diverse accessions resequenced. I realized this is a recent publication while this manuscript has been in review, but it must be addressed in this publication. Specifically, novelty in the current manuscript not presented and how the genomes now compare.

Xia et al 2020 Molecular Plant: The reference genome of tea plant and resequencing of 81 diverse accessions provide insights into genome evolution and adaptation of tea plants

<https://www.cell.com/action/showPdf?pii=S1674-2052%2820%2930134-9>

Author response: We are happy to see the publish of Xia's article on Molecular Plant during the process of our manuscript's critical review. We had read that fantastic paper and discussed in the discussion. The publication of multiple papers related to the similar field showed the critical role of tea genome research.

For plant, extensive experience from simple genome like rice, soybean, and maize have showed that more high quality reference genomes will facilitate the genome, functional genomics and breeding research, this was also true for a complex genome like tea, with the improvement of genome sequencing technology, we hope to see more tea genomes are assembled in the future, the methods we and others have set up will definitely with high reference value.

According to the suggestion, we cited the paper of Xia et al 2020 in line 326. It happened that similar paper published during the review process of another one. We also want to thanks the reviewers again for their rounds of critical review to help us to enhance the innovation and quality of our manuscript. We expect that our article will benefit the tea functional genomics and breeding research in the future.

Reviewer #4 (Remarks to the Author):

Tea plant (*Camelia sinensis*) has a rich, long and complex history that generated the extant populations of the major producing countries such as China and India, However, little is known on about the wild ancestor of this species (M. K. Meegahakumbura et al. 2016; Muditha K. Meegahakumbura et al. 2017; Yao et al. 2012; Kingdom-Ward 1950). Despite its

diploid genome efforts to assemble its genome have been overwhelmed by the complexity of the genome adding the factor that it is self-incompatible making it highly heterozygous factor that increase the difficulty of the assembly (Wei et al. 2018). To this date, tea still a interesting model to study selection dynamics that lead to the local adaptation of the different populations as result of this wide dispersion.

In this manuscript, Xinchao Wang and co-authors achieved an improvement of the genomic reference when compared to the previous reference published in PNAS by Wei et al 2018, by using long read sequencing and the physical interaction provided by Hi-C data. Increasing the N50 size of the scaffolds from less than 1.4 Mb to 143.85 Mb, increasing the amount of core genic annotations by 7.8%, and anchoring around 2.3 Gbp out of 3.26 Gbp to chromosomal level. Also, they attempted to disentangle the domestication history of the crop by sequencing of 134 tea accessions that were grouped in 3 different populations (CSS, C. sinensis var. sinensis; CSA, C. sinensis var. assamica and CSR, C. sinensis-related species). They presented a evidence showing in the form of a PCA that the first two principal components explained 13.08% of the total variance observed between all the sampled individuals. Maximum Likelihood phylogenetic tree that suggested the group CSR as ancestral to the rest of the groups (CSS and CSA).

This manuscript provides a noticeable improvement of *Camelia sinensis* genome resources and also gives a complete insight of the evolution of the Chinese tea accessions and its dispersal across many countries.

Even that the analysis conducted in your manuscript were conducted with high quality and standards, the discussion section can benefit from the argumentation provided in this review.

Q4: for the discussion section I suggest you to consider the limitations of the sampling in regard to the India origin and the exploit the opportunities that the sampling has in terms of the expansion of the origin of two main populations of China tea and its dispersal across the world.

Author response: We appreciate for the constructive comments and suggestions for our manuscript! We had considered the limitations of the sampling in India, it could not rule out other possibilities when more comprehensive samples were involved, so we had weakened the origin results in the discussion. The collection of more India sample in the future will be benefit for the origin and evolution analysis, and we emphasized it in the discussion (lines 322-325).

Major comments

Q5: Main text line 370 to 372. the argument claiming that the breeding of the plants has largely been determined by the environment rather than human behavior is attributed to a lack of severe domestication is not supported by clear evidence. the Supplementary Figs 20 and 21 show changes in population size through out time for the CSA and CSS clusters, however this is not an evidence that the environment created the given tendencies. in contrast Your F4 analysis suggest a strong gene flow between the accessions and is a

evidence that the genetic diversity reduced by any bottleneck may be recovered as result of recombination masking any previous bottleneck.

To improve the discussion and overall interpretation of the Supplementary Fig. 20 I would include the CSR cluster into the Ne analysis to see the moment in which the populations from CSA and CSS change in size respect CSR. If the change of CSR respects the other groups occurred at a point where CSA and CSS were very similar that may be another evidence to suggest the origin of CSA and CSS happened at a similar time point. Also please make clear the X axis of that figure to clearly see the age before present of the time points; the given exponential notation complicates the interpretation.

Author response: Thanks for the valuable suggestion. We agreed with you that a strong gene flow between the accessions and is an evidence that the genetic diversity reduced by any bottleneck may be recovered as result of recombination masking any previous bottleneck. We had deleted the previous speculation of “breeding of tea plants has largely been determined by environmental influences rather than human behavior”, and added the discussion in lines 365-369. “There is no population bottleneck during the history of tea cultivation (Supplementary Material, Supplementary Figs. 20 and 21), this maybe due to the widespread gene flow between different tea that may mask the recent bottlenecks or the tea plants have not experienced strong domestication selection, which also indicated the complex history of tea evolution and domestication.”

We supplied the CSR in the Supplementary Fig. 20, and adjusted the x axis to see the age clearly (Supplementary Fig. 20).

Q6: Main text line 380 to 389. in these lines you describe that the evidence that CSR is ancestral to CSS and CSA make doubtful the observation of a probably independent domestication event in China and India provided by Meegahakumbura et al. 2016. I do not agree with your claim, as not a single accession from India was included in your analysis. Therefore, the possibility of an independent event still valid. What is very interesting about having CSR as ancestor of CSS and CSA is that it may be an evidence of a unique origin of the CSR and CSA clusters that may be dispersed after this event as far as Africa, Georgia or Hawaii.

Author response: Thank you for pointing this out. According to the opinion described in Choi and Purugganan, 2018; Civan et al. 2016, and many similar papers, the domestication analysis should be based on the phylogenetic tree of domesticated gene in a species. For tea, the genomic and functional genomics research was still at the very beginning stage and it was difficult for its domestication analysis. Thus, our manuscript only discussed the evolution of tea. We think it is far from the conclusion for how many times of origin and how many times of domestication with the current data. The opinion of independent domestication events is different to ours of one origin event. It is also possible that had one origin event but multiple domestication ones if you definite domestication as breeding and localization. The CSR may be the

ancestor of CSS and CSA was based the phylogenetic tree.

In the present study, we only have one accession from India (ASM, Supplementary Table 16) clustered to CSA (Supplementary Fig 14). The limitations of the sampling in India could not rule out other possibilities when more comprehensive samples were involved. We did not agree the opinion of three independent domestication event in China and India provided by Meegahakumbura et al. 2016, first, they did not use the outgroup in their phylogenetic tree, one could not get the conclusion of three independent domestication by three clusters without the outgroup. Second, their result was based on the phylogenetic tree of SSR marker, not domesticated gene, and was not suitable to draw any conclusion for domestication.

Q7: Supplementary materials 356 to 364. please include the error values for K clustering in the admixture analysis to confirm that choosing a K=3 is the most likely scenario of clustering and that there are not other more likely clusters.

Author response: Thank you for pointing this out. We had supplied the figure in supplementary material line 354. We run admixture with K values (number of assumed ancestral components) ranging from 1 to 10, and the results shown k=3 had the lowest cv-errors (Supplementary Fig 13).

- Adiconis, X., D. Borges-Rivera, R. Satija, D. S. DeLuca, M. A. Busby, A. M. Berlin, A. Sivachenko, D. A. Thompson, A. Wysoker, T. Fennell, A. Gnirke, N. Pochet, A. Regev, and J. Z. Levin. 2013. 'Comparative analysis of RNA sequencing methods for degraded or low-input samples', *Nature Methods*, 10: 623-9.
- Du, X. M., G. Huang, S. P. He, Z. E. Yang, G. F. Sun, X. F. Ma, N. Li, X. Y. Zhang, J. L. Sun, M. Liu, Y. H. Jia, Z. E. Pan, W. F. Gong, Z. H. Liu, H. Q. Zhu, L. Ma, F. Y. Liu, D. G. Yang, F. Wang, W. Fan, Q. Gong, Z. Peng, L. R. Wang, X. Y. Wang, S. J. Xu, H. H. Shang, C. R. Lu, H. K. Zheng, S. W. Huang, T. Lin, Y. X. Zhu, and F. G. Li. 2018. 'Resequencing of 243 diploid cotton accessions based on an updated A genome identifies the genetic basis of key agronomic traits', *Nature Genetics*, 50: 796-+.
- Duan, N. B., Y. Bai, H. H. Sun, N. Wang, Y. M. Ma, M. J. Li, X. Wang, C. Jiao, N. Legall, L. Y. Mao, S. B. Wan, K. Wang, T. M. He, S. Q. Feng, Z. Y. Zhang, Z. Q. Mao, X. Shen, X. L. Chen, Y. M. Jiang, S. J. Wu, C. M. Yin, S. F. Ge, L. Yang, S. H. Jiang, H. F. Xu, J. X. Liu, D. Y. Wang, C. Z. Qu, Y. C. Wang, W. F. Zuo, L. Xiang, C. Liu, D. Y. Zhang, Y. Gao, Y. M. Xu, K. N. Xu, T. Chao, G. Fazio, H. R. Shu, G. Y. Zhong, L. L. Cheng, Z. J. Fei, and X. S. Chen. 2017. 'Genome re-sequencing reveals the history of apple and supports a two-stage model for fruit enlargement', *Nature Communications*, 8.
- Hussain, M., M. A. Iqbal, B. J. Till, and Mehboob-ur-Rahman. 2018. 'Identification of induced mutations in hexaploid wheat genome using exome capture assay', *Plos One*, 13.
- Jia, J., S. Zhao, X. Kong, Y. Li, G. Zhao, W. He, R. Appels, M. Pfeifer, Y. Tao, X. Zhang, R. Jing, C.

- Zhang, Y. Ma, L. Gao, C. Gao, M. Spannagl, K. F. Mayer, D. Li, S. Pan, F. Zheng, Q. Hu, X. Xia, J. Li, Q. Liang, J. Chen, T. Wicker, C. Gou, H. Kuang, G. He, Y. Luo, B. Keller, Q. Xia, P. Lu, J. Wang, H. Zou, R. Zhang, J. Xu, J. Gao, C. Middleton, Z. Quan, G. Liu, J. Wang, Consortium International Wheat Genome Sequencing, H. Yang, X. Liu, Z. He, L. Mao, and J. Wang. 2013. 'Aegilops tauschii draft genome sequence reveals a gene repertoire for wheat adaptation', *Nature*, 496: 91-5.
- Ling, H. Q., B. Ma, X. L. Shi, H. Liu, L. L. Dong, H. Sun, Y. H. Cao, Q. Gao, S. S. Zheng, Y. Li, Y. Yu, H. L. Du, M. Qi, Y. Li, H. W. Lu, H. Yu, Y. Cui, N. Wang, C. L. Chen, H. L. Wu, Y. Zhao, J. C. Zhang, Y. W. Li, W. J. Zhou, B. R. Zhang, W. J. Hu, M. J. T. van Eijk, J. F. Tang, H. M. A. Witsenboer, S. C. Zhao, Z. S. Li, A. M. Zhang, D. W. Wang, and C. Z. Liang. 2018. 'Genome sequence of the progenitor of wheat A subgenome *Triticum urartu*', *Nature*, 557: 424-+.
- Ling, H. Q., S. Zhao, D. Liu, J. Wang, H. Sun, C. Zhang, H. Fan, D. Li, L. Dong, Y. Tao, C. Gao, H. Wu, Y. Li, Y. Cui, X. Guo, S. Zheng, B. Wang, K. Yu, Q. Liang, W. Yang, X. Lou, J. Chen, M. Feng, J. Jian, X. Zhang, G. Luo, Y. Jiang, J. Liu, Z. Wang, Y. Sha, B. Zhang, H. Wu, D. Tang, Q. Shen, P. Xue, S. Zou, X. Wang, X. Liu, F. Wang, Y. Yang, X. An, Z. Dong, K. Zhang, X. Zhang, M. C. Luo, J. Dvorak, Y. Tong, J. Wang, H. Yang, Z. Li, D. Wang, A. Zhang, and J. Wang. 2013. 'Draft genome of the wheat A-genome progenitor *Triticum urartu*', *Nature*, 496: 87-90.
- Ma, Z. Y., S. P. He, X. F. Wang, J. L. Sun, Y. Zhang, G. Y. Zhang, L. Q. Wu, Z. K. Li, Z. H. Liu, G. F. Sun, Y. Y. Yan, Y. H. Jia, J. Yang, Z. E. Pan, Q. S. Gu, X. Y. Li, Z. W. Sun, P. H. Dai, Z. W. Liu, W. F. Gong, J. H. Wu, M. Wang, H. W. Liu, K. Y. Feng, H. F. Ke, J. D. Wang, H. Y. Lan, G. N. Wang, J. Peng, N. Wang, L. R. Wang, B. Y. Pang, Z. Peng, R. Q. Li, S. L. Tian, and X. M. Du. 2018. 'Resequencing a core collection of upland cotton identifies genomic variation and loci influencing fiber quality and yield', *Nature Genetics*, 50: 803-+.
- Maccaferri, M., N. S. Harris, S. O. Twardziok, R. K. Pasam, H. Gundlach, M. Spannagl, D. Ormanbekova, T. Lux, V. M. Prade, S. G. Milner, A. Himmelbach, M. Mascher, P. Bagnaresi, P. Faccioli, P. Cozzi, M. Lauria, B. Lazzari, A. Stella, A. Manconi, M. Gnocchi, M. Moscatelli, R. Avni, J. Deek, S. Biyiklioglu, E. Frascaroli, S. Corneti, S. Salvi, G. Sonnante, F. Desiderio, C. Mare, C. Crosatti, E. Mica, H. Ozkan, B. Kilian, P. De Vita, D. Marone, R. Joukhadar, E. Mazzucotelli, D. Nigro, A. Gadaleta, S. M. Chao, J. D. Faris, A. T. O. Melo, M. Pumphrey, N. Pecchioni, L. Milanesi, K. Wiebe, J. Ens, R. P. MacLachlan, J. M. Clarke, A. G. Sharpe, C. S. Koh, K. Y. H. Liang, G. J. Taylor, R. Knox, H. Budak, A. M. Mastrangelo, S. S. Xu, N. Stein, I. Hale, A. Distelfeld, M. J. Hayden, R. Tuberosa, S. Walkowiak, K. F. X. Mayer, A. Ceriotti, C. J. Pozniak, and L. Cattivelli. 2019. 'Durum wheat genome highlights past domestication signatures and future improvement targets', *Nature Genetics*, 51: 885-+.
- Mayer, K. F. X., J. Rogers, J. Dolezel, C. Pozniak, K. Eversole, C. Feuillet, B. Gill, B. Friebe, A. J. Lukaszewski, P. Sourdille, T. R. Endo, J. Dolezel, M. Kubalakova, J. Cihalikova, Z. Dubska, J. Vrana, R. Sperkova, H. Simkova, J. Rogers, M. Febrer, L. Clissold, K. McLay, K. Singh, P. Chhuneja, N. K. Singh, J. Khurana, E. Akhunov, F. Choulet, P. Sourdille, C. Feuillet, A. Alberti, V. Barbe, P. Wincker, H. Kanamori, F. Kobayashi, T. Itoh, T. Matsumoto, H. Sakai, T. Tanaka, J. Z. Wu, Y. Ogihara, H. Handa, C. Pozniak, P. R. Maclachlan, A. Sharpe, D. Klassen, D. Edwards, J. Batley, O. A. Olsen, S. R. Sandve, S. Lien, B. Steuernagel, B. Wulff, M. Caccamo, S. Ayling, R. H. Ramirez-Gonzalez, B. J. Clavijo, B. Steuernagel, J. Wright, M.

- Pfeifer, M. Spannagl, K. F. X. Mayer, M. M. Martis, E. Akhunov, F. Choulet, K. F. X. Mayer, M. Mascher, J. Chapman, J. A. Poland, U. Scholz, K. Barry, R. Waugh, D. S. Rokhsar, G. J. Muehlbauer, N. Stein, H. Gundlach, M. Zytnicki, V. Jamilloux, H. Quesneville, T. Wicker, K. F. X. Mayer, P. Faccioli, M. Colaiacovo, M. Pfeifer, A. M. Stanca, H. Budak, L. Cattivelli, N. Glover, M. M. Martis, F. Choulet, C. Feuillet, K. F. X. Mayer, M. Pfeifer, L. Pingault, K. F. X. Mayer, E. Paux, M. Spannagl, S. Sharma, K. F. X. Mayer, C. Pozniak, R. Appels, M. Bellgard, B. Chapman, M. Pfeifer, M. Pfeifer, S. R. Sandve, T. Nussbaumer, K. C. Bader, F. Choulet, C. Feuillet, K. F. X. Mayer, E. Akhunov, E. Paux, H. Rimbart, S. C. Wang, J. A. Poland, R. Knox, A. Kilian, C. Pozniak, M. Alaux, F. Alfama, L. Couderc, V. Jamilloux, N. Guilhot, C. Viseux, M. Loaec, H. Quesneville, J. Rogers, J. Dolezel, K. Eversole, C. Feuillet, B. Keller, K. F. X. Mayer, O. A. Olsen, S. Praud, and Iwgc. 2014. 'A chromosome-based draft sequence of the hexaploid bread wheat (*Triticum aestivum*) genome', *Science*, 345.
- Nam, D. K., S. Lee, G. L. Zhou, X. H. Cao, C. Wang, T. Clark, J. J. Chen, J. D. Rowley, and S. M. Wang. 2002. 'Oligo(dT) primer generates a high frequency of truncated cDNAs through internal poly(A) priming during reverse transcription', *Proceedings of the National Academy of Sciences of the United States of America*, 99: 6152-56.
- Salojärvi, J., O. P. Smolander, K. Nieminen, S. Rajaraman, O. Safronov, P. Safdari, A. Lamminmaki, J. Immanen, T. Lan, J. Tanskanen, P. Rastas, A. Amiryousefi, B. Jayaprakash, J. I. Kammonen, R. Hagqvist, G. Eswaran, V. H. Ahonen, J. A. Serra, F. O. Asiegbu, J. de Dios Barajas-Lopez, D. Blande, O. Blokhina, T. Blomster, S. Broholm, M. Brosche, F. Cui, C. Dardick, S. E. Ehonen, P. Elomaa, S. Escamez, K. V. Fagerstedt, H. Fujii, A. Gauthier, P. J. Gollan, P. Halimaa, P. I. Heino, K. Himanen, C. Hollender, S. Kangasjarvi, L. Kauppinen, C. T. Kelleher, S. Kontunen-Soppela, J. P. Koskinen, A. Kovalchuk, S. O. Karenlampi, A. K. Karkonen, K. J. Lim, J. Leppala, L. Macpherson, J. Mikola, K. Mouhu, A. P. Mahonen, U. Niinemets, E. Oksanen, K. Overmyer, E. T. Palva, L. Pazouki, V. Pennanen, T. Puhakainen, P. Poczai, Bjhm Possen, M. Punkkinen, M. M. Rahikainen, M. Rousi, R. Ruonala, C. van der Schoot, A. Shapiguzov, M. Sierla, T. P. Sipila, S. Sutela, T. H. Teeri, A. I. Tervahauta, A. Vaattovaara, J. Vahala, L. Vetchinnikova, A. Welling, M. Wrzaczek, E. Xu, L. G. Paulin, A. H. Schulman, M. Lascoux, V. A. Albert, P. Auvinen, Y. Helariutta, and J. Kangasjarvi. 2017. 'Genome sequencing and population genomic analyses provide insights into the adaptive landscape of silver birch', *Nature Genetics*, 49: 904-12.
- Varshney, R. K., R. K. Saxena, H. D. Upadhyaya, A. W. Khan, Y. Yu, C. Kim, A. Rathore, D. Kim, J. Kim, S. An, V. Kumar, G. Anuradha, K. N. Yamini, W. Zhang, S. Muniswamy, J. S. Kim, R. V. Penmetsa, E. von Wettberg, and S. K. Datta. 2017. 'Whole-genome resequencing of 292 pigeonpea accessions identifies genomic regions associated with domestication and agronomic traits', *Nature Genetics*, 49: 1082-88.
- Wang, W. S., R. Mauleon, Z. Q. Hu, D. Chebotarov, S. S. Tai, Z. C. Wu, M. Li, T. Q. Zheng, R. R. Fuentes, F. Zhang, L. Mansueto, D. Copetti, M. Sanciangco, K. C. Palis, J. L. Xu, C. Sun, B. Y. Fu, H. L. Zhang, Y. M. Gao, X. Q. Zhao, F. Shen, X. Cui, H. Yu, Z. C. Li, M. L. Chen, J. Detras, Y. L. Zhou, X. Y. Zhang, Y. Zhao, D. Kudrna, C. C. Wang, R. Li, B. Jia, J. Y. Lu, X. C. He, Z. T. Dong, J. B. Xu, Y. H. Li, M. Wang, J. X. Shi, J. Li, D. B. Zhang, S. Lee, W. S. Hu, A. Poliakov, I. Dubchak, V. J. Ulat, F. N. Borja, J. R. Mendoza, J. Ali, J. Li, Q. Gao, Y. C. Niu, Z. Yue, M. E. B. Naredo, J. Talag, X. Q. Wang, J. J. Li, X. D. Fang, Y. Yin, J. C. Glaszmann, J. W. Zhang, J. Y. Li, R. S. Hamilton, R. A. Wing, J. Ruan, G. Y. Zhang, C. C.

Wei, N. Alexandrov, K. L. McNally, Z. K. Li, and H. Leung. 2018. 'Genomic variation in 3,010 diverse accessions of Asian cultivated rice', *Nature*, 557: 43-+.

REVIEWERS' COMMENTS:

Reviewer #2 (Remarks to the Author):

Many thanks for your comprehensive response to my questions. These minor issues have now been resolved and I have no further questions. Overall, the manuscript is much improved and represents a significant understating of the genomics and evolution of this important species.

Reviewer #4 (Remarks to the Author):

Dear editor

I have finished reviewing the manuscript entitled "Population sequencing enhances understanding of tea plant evolution " submitted to nature communications by Wang et al.

This work attempts to investigate the domestication history of the cultivated tea plants (*Camellia sinensis*); by generating a improved genomic reference, sequencing and phylogenetic analysis of 134 tea accessions and the identification of divergent selection on 3 different populations (CSS, *C. sinensis* var. *sinensis*; CSA, *C. sinensis* var. *assamica* and CSR, *C. sinensis*-related species) in the form of positive selection.

In this manuscript the authors explore at a genome wide level the phylogenetic relationship of a wide sampling of tea populations dispersed mainly in China but also some populations as far as Africa, Georgia or Hawaii. They also identify one population to be ancestral to all the sampled accessions and at the same described the complex demographics of the populations.

During 4 reviews of this manuscript the authors have polished the arguments supported by their evidences giving an interesting perspective on the understanding of the tea plant evolution. Therefore, I recommend this manuscript to be published in Nature Communications after attending a couple of minor comments.

Thank you again for the opportunity of reviewing this interesting manuscript.

Best wishes

Miguel Vallebuena

[Editor: In remark to Editor section, Reviewer #4 points out that the discussion would be enriched by describing the extent of admixing between the Chinese accessions using the Africa, Georgia or Hawaii as a contrast. In addition, (s)he recommends to shade the argument "There are no population bottleneck" to "Our results show no signs of a clear bottleneck occurring in the last 20K years" in line 365.]

REVIEWERS' COMMENTS:

Reviewer #2 (Remarks to the Author):

Many thanks for your comprehensive response to my questions. These minor issues have now been resolved and I have no further questions. Overall, the manuscript is much improved and represents a significant understating of the genomics and evolution of this important species.

Author response: Thanks for your help in improving our manuscript.

Reviewer #4 (Remarks to the Author):

Dear editor

I have finished reviewing the manuscript entitled "Population sequencing enhances understanding of tea plant evolution" submitted to nature communications by Wang et al.

This work attempts to investigate the domestication history of the cultivated tea plants (*Camellia sinensis*); by generating a improved genomic reference, sequencing and phylogenetic analysis of 134 tea accessions and the identification of divergent selection on 3 different populations (CSS, *C. sinensis* var. *sinensis*; CSA, *C. sinensis* var. *assamica* and CSR, *C. sinensis*-related species) in the form of positive selection.

In this manuscript the authors explore at a genome wide level the phylogenetic relationship of a wide sampling of tea populations dispersed mainly in China but also some populations as far as Africa, Georgia or Hawaii. They also identify one population to be ancestral to all the sampled accessions and at the same described the complex demographics of the populations.

During 4 reviews of this manuscript the authors have polished the arguments supported by their evidences giving an interesting perspective on the understanding of the tea plant evolution. Therefore, I recommend this manuscript to be published in Nature Communications after attending a couple of minor comments.

Thank you again for the opportunity of reviewing this interesting manuscript.

Best wishes

Miguel Vallebueno

[Editor: In remark to Editor section, Reviewer #4 points out that the discussion would be enriched by describing the extent of admixing between the Chinese accessions using the

Africa, Georgia or Hawaii as a contrast. In addition, (s)he recommends to shade the argument “There are no population bottleneck” to “Our results show no signs of a clear bottleneck occurring in the last 20K years” in line 365.]

Author response: According to the valuable suggestion from reviewer 4, we have replaced “There are no population bottleneck” with “Our results show no signs of a clear bottleneck occurring in the last 20,000 years” in line 351, we think that the suggested expression about this opinion is much better now.

The suggestion to enrich the discussion related to the extent of tea admixture was a good point, and actually we have done similar analysis (Supplementary Note 4) but the result appeared trite and insignificant, thus we did not put many words on it. According to the reviewer’s suggestion, We further selected the groups which only contain the Chinese samples (324 groups, 99.38% groups contain gene flow), groups contain only the samples from outside of China (42 groups, all groups contain gene flow), and mixed of all Chinese samples and samples from outside of China (634 groups, 98.11% groups contain gene flow) in F4-test results (Supplementary Data 10). There is no significant difference between Chinese samples and the samples from outside of China (fisher.test, P-value >0.05). Most of tea samples from outside of China are spreading from China in the past more than 100 years ago, and for this opinion, though most of the related papers were published in Chinese journals on culture or economics, some of them have been validated by molecular techniques such as the tea breeding history in Africa (Wambulwa et al, 2016). The extent of admixing between the Chinese accessions using the Africa, Georgia or Hawaii as a contrast don’t help to improve the manuscript, so we do not supply this information in the discussion.

Wambulwa, M. C., et al. (2016). Insights into the genetic relationships and breeding patterns of the African tea germplasm based on nSSR Markers and cpDNA sequences. *Front Plant Sci* 7: 1244.